# Retrospective Study of the Seroprevalence of HIV, HCV, and HBV in Blood Donors at a Blood Bank of Western Mexico

**DOI:** 10.3390/pathogens10070878

**Published:** 2021-07-11

**Authors:** José de Jesús Guerrero-García, Alejandra Guadalupe Zúñiga-Magaña, Juan Carlos Barrera-De León, Rafael Magaña-Duarte, Daniel Ortuño-Sahagún

**Affiliations:** 1Banco de Sangre Central, UMAE Hospital de Especialidades CMNO, Instituto Mexicano del Seguro Social (IMSS), Guadalajara 44340, Jalisco, Mexico; rafael.magana@imss.gob.mx; 2Instituto de Investigación en Ciencias Biomédicas (IICB), CUCS, Universidad de Guadalajara, Guadalajara 44340, Jalisco, Mexico; alejandra.zmagana@alumnos.udg.mx; 3Centro Médico Nacional de Occidente (CMNO), Unidad Médica de Alta Especialidad (UMAE), Hospital de Pediatría (HP), Instituto Mexicano del Seguro Social (IMSS), Guadalajara 44340, Jalisco, Mexico; jcbarrer@hotmail.com; 4Centro Universitario de Ciencias de la Salud, Universidad de Guadalajara, Guadalajara 44340, Jalisco, Mexico

**Keywords:** seroprevalence, TTIs, blood donors, safe blood, Mexico

## Abstract

Obtaining blood which is safe for transfusions is one of the principal challenges in the health systems of developing countries. Supply of contaminated blood increases morbidity, mortality, and the costs of patient care. In Mexico, serological screening is mandatory, but only a few of the main blood banks routinely perform a nucleic acid test (NAT). Data from 80,391 blood donations processed between August 2018 and December 2019 at the Central Blood Bank of the Western National Medical Center of the Mexican Social Security Institute (IMSS) were analyzed. All donors were screened for serological markers and NAT was performed. Reactive donors were followed-up to confirm their results. The number of reactive donors and seroprevalence rates for HIV, HCV, and HBV were 152 (18.91/10,000), 385 (47.89/10,000), and 181 (22.51/10,000), respectively; however, these rates decreased when NAT-confirmed reactive results were considered. Male donors were found to have a higher seroprevalence than females, and younger donors higher than older donors. The present study shows that HIV, HCV, and HBV seroprevalence in blood donors in Western Mexico is low. We propose that Mexico should establish future strategies, including pathogen reduction technologies (PRTs), in order to improve blood safety and reduce transfusion-transmissible infections (TTIs).

## 1. Introduction

Blood transfusion is a therapeutic intervention that can improve or even save lives. Obtaining safe blood and its components, however, is one of the greatest challenges in poorly developed and developing countries. Supply of contaminated blood not only increases morbidity and mortality, but also increases the costs of patient care within the health system. In this sense, the costs of screening for and researching infectious agents which are potentially transmitted by blood constitutes a profitable investment for all countries [1].

Global blood donations average 118.5 million per year, and 60% are donated in low- and middle-income countries, where 84% of the world’s population lives [2]. Blood donations must be screened for infectious agents, including human immunodeficiency virus (HIV), hepatitis C virus (HCV), and hepatitis B virus (HBV), according to recommendations by the World Health Organization (WHO) [3]. Since the discovery of HIV and the increase in blood transfusion-related cases in the early 1980s, research aimed at improving blood safety has enabled the development of advanced technology for donor screening, and the risk of transfusion-transmissible infections (TTIs) has been markedly reduced [4].

However, TTIs are still a serious problem in areas of public health and compromise blood availability and safety, especially in developing countries where financial resources and blood bank infrastructure are limited [5]. To decrease the risk of TTIs and ensure access to safe blood and its components, WHO issued a series of recommendations with the aim of establishing national systems and legislation that regulate the effective collection of donations, routine screening for TTIs, the implementation of quality systems, and the rational use of blood products [3].

The global distribution of TTIs varies according to the region of the world and even between neighboring countries, a fact which may be related to the economic status of each country. Current prevalence of TTIs in high-income countries is lower (HIV 0.001%, HBV 0.01%, HCV 0.06%) than in low- (HIV 0.70%, HBV 2.81%, HCV 1.00%) and middle-income countries (HIV 0.10 to 0.19%, HBV 0.29 to 1.96%, HCV 0.18 to 0.38%), primarily because high-income countries have well-defined and mandatory blood donor selection programs [6]. In Mexico, the estimated national average seroprevalence of HIV, HBV, and HCV in 2015 was 0.17%, 0.15%, and 0.48%, respectively [5]. However, there are differing cultural, demographic, geographic, and economic conditions between the different regions of the country which may influence the local seroprevalence of TTIs. 

The development in the early 1990s of molecular biology technologies, such as polymerase chain reaction (PCR) and transcription-mediated amplification (TMA), allowed for the implementation of nucleic acid testing (NAT) for blood screening in order to detect the genetic material of HIV, HBV, and HCV, which has a paramount relevance for the reduction of the residual risk of window periods [7]. Although only serological screening is mandatory in Mexico, a few of the main blood banks in the country routinely perform a nucleic acid test (NAT) to increase the safe blood supply. The Central Blood Bank of the Specialties Hospital of the Western National Medical Center of the Mexican Social Security Institute receives an average of 68,000 donations per year, making it the bank with the second highest blood collection and blood component production in the country. Whole blood donations were serologically screened for TTIs and NAT was performed using the Procleix Ultrio Assay (GRIFOLS), as described in Materials and Methods. The aim of this study was to determine the seroprevalence of HIV, HBV, and HCV in blood donors from Western Mexico who attended the Central Blood Bank between August 2018 and December 2019.

## 2. Results

The demographic data of donors are presented in Table 1. The data of 80,391 voluntary blood donors, including gender, age, and location/area, were analyzed in this study; however, only 78,276 donors left a record of their age. Men made up 67.2% of all donors and 32.8% were women (ratio 2:1). According to the area of residence, 77.8% of donors were from the metropolitan area of Guadalajara and 22.2% were from adjacent regions. The average age of donors was 33.9 ± 10.6 years. When donors were stratified by age range, we found that younger donors (18–30 years) made the highest frequency of donations (43.4%), compared with older donors (51–65 years), who had the lowest frequency (7.7%). The two most frequent ABO blood group were O with 59.9% and A with 28.5%. A total of 94.2% of donors were Rh (D antigen) positive and 5.8% were Rh negative. Based on this data, younger males from the metropolitan area with blood group O who are Rh positive are the most frequent blood donors.

### Seroprevalence of HIV, HCV, and HBV in Blood Donors

From the serological screening of all donations, the number of seroreactive donors and rates, expressed in cases per 10,000, of HIV, HCV, and HBV were 152 (18.91/10,000), 385 (47.89/10,000), and 181 (22.51/10,000), respectively (Table 2). When donors were classified by gender, the results showed that male donors had a higher seroprevalence (13.43/10,000 HIV, 29.85/10,000 HCV, and 12.07/10,000 HBV) than females (5.47/10,000 HIV, 18.04/10,000 HCV, and 10.45/10,000 HBV) for the three TTIs in male/female ratios of 3:1 for HIV, 5:3 for HCV, and 5:4 for HBV (Table 3). When classified by age range, seroprevalence was higher in younger donors and decreased in older donors, going from 8.81/10,000 HIV, 15.84/10,000 HCV, and 9.96/10,000 HBV in younger donors to 0.51/10,000 HIV, 6.52/10,000 HCV, and 1.79/10,000 HBV in older donors (Table 3). From the analysis of these categorical variables with respect to the screening results, we found that HCV (*p* = 0.041) and HBV (*p* = 0.000) are related to being male, but with a low intensity association. This association may be attributed to the fact that the higher number of reactive cases in males is related to the higher proportion of male donors compared with female donors (2:1), and not to gender per se. As for the other variables, no other association was found. When seroprevalence was classified by educational level, we observed that junior high school, high school, and people with school degrees were the groups with higher seroprevalence of the three TTIs. Additionally, “employee” was the occupational group with the highest seroprevalence of the three TTIs (Appendix A).

The data were then analyzed to determine the number of window periods. A window period is defined as the period between the time of infection and the production of detectable markers in serum. Window periods were detected in the screening blood donors by performing NAT and the respective discriminatory test for those samples with negative serological screening. In all blood donors, the number of cases detected for window periods was seven (0.87/10,000), of which one was HIV (0.12/10,000), four were HCV (0.50/10,000), and two were HBV (0.25/10,000) (Table 4). Interestingly, when we crossed the results of these window periods with the results of the reactive serological screening, we found that two HCV window periods were, in addition, HIV-reactive donors (1073.8 S/CO and 627.98 S/CO, respectively). Although the number of window periods is very low, as we expected, these donations represent at least three discarded blood components per donation (red cell concentrate, platelet concentrate, and fresh frozen plasma or cryoprecipitate plasma), which would have posed a risk of transmission of any of the TTIs to transfused patients, caused by blood banks that do not perform NAT and supply blood components that have only undergone serological screening.

For the purpose of this study, in order to detect true active infections and eliminate false reactive results inherent to the technology used [8,9], and based on the detection of viral RNA or DNA, repeatedly reactive samples—both in serological screening and NAT, plus detected window periods—were considered to be confirmed-positive reactive results. In this regard, the number of cases and seroprevalences of TTIs were reduced to 33 (4.10/10,000) for HIV, 49 (6.10/10,000) for HCV, and 22 (2.74/10,000) for HBV (Table 4). 

Based on this analysis and the apparently high number of false reactive results, the reactive serological cut-off points were analyzed as follows. First, according to national regulations, reactive and indeterminate donors must be notified with a minimum of three attempts to locate in order to obtain a second sample. Of the 33 confirmed reactive results for HIV, 49 for HCV, and 22 for HBV in the three TTIs (Table 4), only 7/33 (21.2%) for HIV, 10/49 (20.4%) for HCV, and 2/22 (9%) for HBV were collected and found to be positive in the respective confirmatory test. 

Second, of the 152 initial HIV seroreactive results (Table 2), the 32 seroreactives with NAT-reactive screening (Table 4) had cutoff values from 188.9 to 1514.2 (S/CO). In contrast, of the remaining 120 NAT-negative HIV seroreactives, 100 corresponded to cutoff values ≤3.0 (S/CO), 13 to cutoff values between 3.0 and 10.0 (S/CO), and the last 7 to cutoff values >10.0 (S/CO). In the same line, of these 120 HIV seroreactive donors, only 38 donors were located, of which 37 were confirmed as negative in this second sample and only 1 donor was positive in the respective confirmatory test. Third, of the 385 initial HCV seroreactives (Table 2), 43/45 seroreactives with NAT-reactive screening (Table 4) had cutoffs >10.0 (S/CO), and of the remaining 340 seroreactives with NAT-negative result, only 3 had cutoffs >10.0 (S/CO). Of the latter, only 90 were located and none were confirmed positive with their respective confirmatory test. Finally, regarding the 181 initial seroreactive HBVs (Table 2), 20 were NAT-reactive (cut-offs >800 S/CO) (Table 4), and 161 were NAT-negative. Of these 161 NAT-negative seroreactives, only 33 donors were located and 12 had cut-offs >10.0 (S/CO), and only 4 were positive by confirmatory testing. Despite the low follow-up rate, it is clear that the probability of confirming an active infection, at the time of donation, is higher in those samples with cut-offs >10.0 (S/CO).

## 3. Discussion

In the present study, we analyzed the data of serological and NAT screenings for HIV, HCV, and HBV to determine the seroprevalence of these TTIs among blood donors that attended the Central Blood Bank of the National Western Medical Center of IMSS in Mexico between August 2018 and December 2019, as well as assessing the prevalence of window periods through the comparison of reactive NAT results with their respective negative serological results.

According to the World Bank, Mexico is an upper middle-income country [10] and has a complex health system similar to other countries [11], with a public sector made up of several programs and health institutions with their own organizational structures, regulations, and financial resources, in addition to the private sector. The blood supply is maintained by more than 550 blood banks located around the country. The IMSS is the largest national health institution and provides 47% of the blood components used by public institutions [5]. The Central Blood Bank of the National Western Medical Center receives blood donations mainly from the state of Jalisco, in addition to other states in Western Mexico. This blood bank has an average of 68,000 donations per year, making it the bank with the second highest blood collection and blood component production in the country. Thus, the results presented here primarily represent the profile of the western population of the country (principally the state of Jalisco) and can be used to implement and establish health policies on safe blood control. 

Our results show that there was a low seroprevalence of the three TTIs in the first screening of all donations (18.91/10,000 for HIV, 47.89/10,000 for HCV, and 22.51/10,000 for HBV) (Table 2), which is in line with the data from upper middle-income countries, as reported by WHO, but well above high-income countries [6]. When we classified seroprevalences by age groups, we observed that the seroprevalence of TTIs is higher in younger donors, as in other middle-income countries [6,12]. Moreover, as we expected, male donors have a markedly higher HIV and HCV seroprevalence than females (3:1 ratio for HIV and 5:3 for HCV), but not as marked for HBV (5:4), in addition to a low-intensity association between donor sex and HCV and HBV screening outcome. These data are consistent with the fact that two-thirds of Central Blood Bank donors were male (67.2% male vs. 32.8% female), as was the overall ratio of male to female donors [6,13], suggesting that the results, with respect to gender, may be attributable to the gender ratio in the population. It is important to note that one of the reasons for the lower frequency of female donors is the deferral of pregnant women or is due to the lactation period.

In a recent study involving three large United States blood centers and around 15 million donations between 2011 and 2012, Dodd et al. reported rates for HIV, HCV, and HBV that were 67, 24, and 30 times lower than our results (0.282/10,000, 2.007/10,000, and 0.757/10,000), respectively [13]. These discrepancies are similar to those of other studies, such as the one conducted in the American Red Cross in 2015–2016 for the three TTIs discussed here [14], or even in other upper middle-income countries such as Germany in 2008–2010 [15]. The marked difference in the prevalence of TTIs between high- and middle-income countries suggests that geographical, cultural, and economic differences [14,16,17,18,19] directly influence the prevalence of TTIs among the blood donor population.

In Mexico, the selection and screening of blood donors is established as mandatory by national regulations, although only serological testing is considered mandatory with no obligation to perform NAT [20]. However, the Central Blood Bank performs both tests on all donations to reduce the residual risk inherent to window periods, and is one of the first blood banks to perform NAT in the country, along with other banks in Mexico City and the state of Nuevo León, since 2007–2008 [5]. It is relevant that Dodd et al. introduced the concept of confirmed-positive seroreactive donation samples, which includes any sample with a NAT-reactive result, with or without a reactive serology result, in order to compare different surveillance systems. In this line, when we applied this concept to our study, the analysis showed a decrease in HIV (4.10/10,000), HCV (6.10/10,000), and HBV (2.74/10,000) seroprevalences (Table 4) that corresponded to a 5-, 8-, and 8-fold reduction compared with our reactive serological results alone, respectively. Interestingly, with the application of this concept, it is observed that HIV and HCV seroprevalences are close to the limits corresponding to high-income countries, while HBV seroprevalence is within the limits corresponding to the last ones.

As part of a major effort to increase blood safety, starting from the year 2021, IMSS will make NAT an internal policy, in addition to the corresponding serological screening, for all blood donations collected in its blood banks; the present work provides important evidence demonstrating the advantages in terms of blood safety when institutions invest in the combination of serological screening and NAT application. In addition, a total of 7 TTI window periods were detected among the 80,391 blood donations, or 0.87 per 10,000 donations. In addition, two of the window periods detected for HCV corresponded to two cases of co-infection between HIV and HCV. It is important to note that by complying only with the provisions of NOM-253-SSA1-2012 and without performing NAT screening, these seven window periods would not have been detected, with the risk that patients transfused with these blood components could acquire any of the TTIs, which would result in an increase in morbidity and mortality as well as in a significant increase in the cost of care for new patients with these infectious-contagious diseases.

Serological screening and NAT are two important reactive measures, which have improved the safety and quality of blood in blood banks in the last decades. However, to further increase blood safety and reduce the residual risk inherent in the supply and transfusion of blood components, it is necessary to implement proactive strategies, such as the use of pathogen reduction technology (PRTs) [21]. PRTs consist of technologies that must be implemented in blood banks, such as the use of UV or some chemicals with the purpose of inactivating a broad variety of pathogens, such as viruses, bacteria, and parasites, in blood components before transfusion. This PRT reduces the risk of TTIs through several mechanisms at the protein, structural, or nucleic acid levels [22], and are used in more than two dozen countries [21]. Currently, the Central Blood Bank does not use PRT. In fact, in Mexico, only a few blood banks use PRTs, so it remains necessary to sensitize institutions to the advantages of investing in the use of PRTs with respect to the reduction of TTI infections and thus the cost of patient care.

Finally, despite the difficulties in following up reactive donors, as most donors did not attend the appointment or were impossible to locate, the results obtained from the confirmatory tests performed demonstrate that there is a high number of false reactive results in serological screening that increase seroprevalences, suggesting that in order to produce more reliable results, it is necessary to consider NAT results together with serological screening. It is also crucial to seek strategies and policies that improve the follow-up of reactive blood donors to improve blood safety and decrease our seroprevalence of these TTIs to the levels of high-income countries.

One of the limitations of this study was that the population studied belongs to Western Mexico, primarily the state of Jalisco, and there is not enough information about the rest of the country; of the information that exists, not all of it is current. For example, according to the few existing studies, HCV seroprevalence is higher in the states of Veracruz (1.1%, 2009) (Gulf of Mexico) [23] and Guanajuato (0.87%, 2020) (north-central) [24], compared with the seroprevalence reported in our study (0.48%, Table 2) and that of Yucatán (0.44%, 2006) (south-east) [25]. Therefore, a more robust and comprehensive study with populations from different localities of the country is needed to provide an overall picture of the seroprevalences of these TTIs in Mexico. Another limitation was the low number of second samples obtained in donor follow-up, which translated into a lower number of confirmed TTI cases. This is due to two main reasons: firstly, that these donors provided false personal information for tracing, or secondly, that they openly refused to attend their appointment and receive the results of their serology screening, despite the advantage of knowing in a timely manner if they have a TTI. Currently, in Mexico, there are no policies oriented to blood donation education, which could encourage the open population to donate altruistically, thus increasing the collection of safe blood. Furthermore, there is a limitation regarding the false positive rate related to the screening technology used [9,26,27], which has been reported to be as high as 10.5% for HIV in the case of samples with an S/CO >1, but with low reactivity [28]. These facts reduce the number of positive-confirmed donors detected as reactive in serological screening.

Based on the above, there are two immediate requirements to improve blood safety in Mexico. Firstly, to expand the education or promotion of the culture of donation that has not been firmly established, in which it is a priority to emphasize that, in addition to saving lives, blood donation allows for the timely detection of diseases such as HIV, HCV, and HBV in blood donors, which in turn allows the individual to access treatment in a timely manner. Secondly, it is necessary that the national regulation on blood donation includes as mandatory the use of NAT to reduce the risk of TTIs contagion, which can be considered a long-term investment, since the high costs of managing patients infected with HIV, HCV, and HBV by transfusion of contaminated blood from donors in window periods can exceed USD 5,000, in the case of HIV, per year and per patient [29], and are mainly financed by the public sector.

In summary, the present study shows that the seroprevalence of HIV, HCV, and HBV in blood donors in Western Mexico is low, and corresponds to the rates of upper-middle income countries reported by WHO, but closer to high-income countries, such as the United States or Germany, when blood screening includes NAT. To understand these differences, it is necessary to analyze several factors, such as the cultural, demographic, geographic, and economic conditions of our country. On this basis, we propose that: (1) Mexico has to establish future strategies to increase blood safety and improve the supply of safe blood components, including the use of PRT; (2) NAT must be included as a mandatory test in donor screening, in addition to the determination of serological markers; and (3) the creation of a national database of reactive blood donors, accessible and added to by all blood banks, is necessary. There is still a long way to go to reach first-world blood safety levels in Mexico.

## 4. Materials and Methods

### 4.1. Study Population

A retrospective study was conducted at the Central Blood Bank of the Specialties Hospital of the Western National Medical Center, Mexican Institute of Social Security in Mexico. The data of all voluntary blood donors (80,391) received between August 2018 and December 2019 at the Central Blood Bank or one of their collection centers were included and analyzed. Whole blood obtained at collection centers was sent to the Central Blood Bank for processing and the procurement of its components. Blood group typing and serological and NAT screening was performed in the Laboratory of the Central Blood Bank. The study was conducted in accordance with the ethical guidelines of the 2013 Declaration of Helsinki and was approved by the Ethical and Investigation Committees (R-2020-1301-095) of the Specialties Hospital of Western National Medical Center, Mexican Social Security Institute, in Mexico.

### 4.2. Blood Group Typing

The ABO and Rh (D antigen) blood groups of all donors were determined by: (1) manual forward blood typing by manual tube agglutination with anti-A, anti-B, anti-AB, and anti-Rh test serums; and (2) automated forward and reverse blood typing with Erytra Automated System (GRIFOLS), according to the manufacturer’s instruction. A1 and A2 blood group subtypes were determined using anti-A1 and anti-H for the detection of lectins.

### 4.3. Serological and NAT Screening for HIV, HCV, and HBV

Donor samples were obtained at the time of donation and then screened for serological HIV p24 antigen and anti-HIV-1 (group M and group O) and -HIV-2 antibodies; hepatitis B surface antigen (HBsAg); and anti-HCV antibodies, according to national and institutional regulations (NOM-253-SSA1-2012) [20], and for NAT amplification at the same time.

The determination of serological screening was performed using the automated macroparticle chemiluminescent immunoassay platform ABBOTT Architect i4000 with Architect System HIV Ag/Ab Combo, HBsAg Qualitative II, and Anti-HCV kits. When a sample was “Reactive” (≥1.00 S/CO) or “Undetermined” (0.90 to 0.99 S/CO) for HIV, HCV, or HBV, its plasma or platelet concentrate was sampled and the analysis was repeated in both samples at the same time (Figure 1). 

All donors were also screened for HIV, HCV, and HBV via NAT amplification using the Procleix Ultrio Assay (GRIFOLS) at the same time as their screening for serological markers. When a sample was “Reactive” (≥1.00 S/CO) in NAT amplification, the same sample was analyzed using the Procleix HIV, HBV, and HCV Discriminatory Assay (GRIFOLS), according to the manufacturer’s instruction.

All blood components from reactive or undetermined donors were immediately discarded, according to national and institutional regulations (NOM-253-SSA1-2012) [20].

### 4.4. Reactive Donor’s Follow-up and Confirmatory Analysis

If a reactive or indeterminate result was confirmed in the first serological or NAT screening, at least three attempts were made to locate the donor to obtain a second sample. If donors provided this second sample, screening was performed and, when reactivity was confirmed, it was further analyzed with the corresponding confirmatory test (Figure 2). The confirmatory test platforms used were: (1) New LAV Blot I Assay #72251 BIO-RAD, for HIV; (2) Architect HBsAg Qualitative II Confirmatory, for HBV; and (3) Procleix HCV Discriminatory Assay (GRIFOLS), for HCV. All samples and tests were processed according to the manufacturer’s instruction.

### 4.5. Statistics

Data were processed and analyzed using Microsoft Excel and IBM SPSS Statistics V25.0. To analyze the data, seroprevalences were expressed as rates per 10,000 with 95% confidence intervals (CI) and associations with the categorical variables of gender and age-group were analyzed using Chi-square, Cramer’s V and Lambda tests. Statistical significance was considered when p < 0.05.

## Figures and Tables

**Figure 1 pathogens-10-00878-f001:**
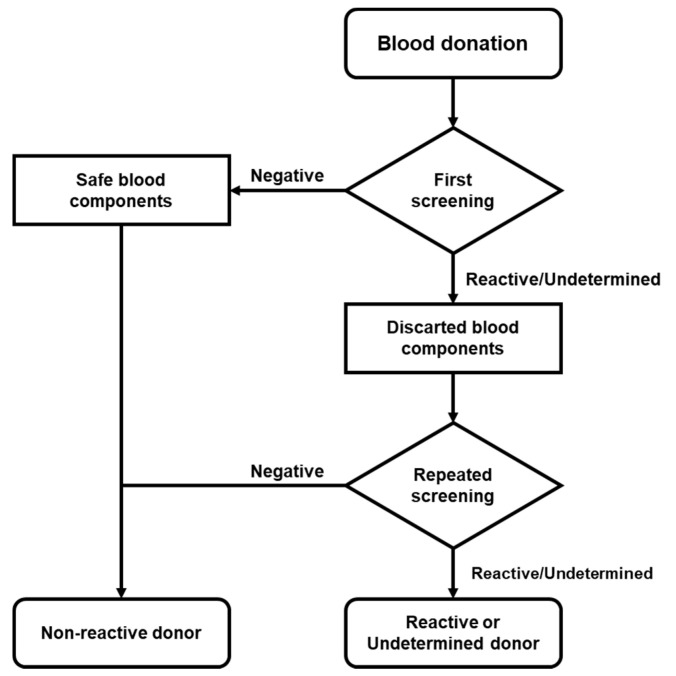
Flow chart of donor screening. Serological screening was performed in accordance with national regulations. When the reactive/undetermined screening was repeated and the donor was considered reactive or indeterminate, follow-up was deemed necessary.

**Figure 2 pathogens-10-00878-f002:**
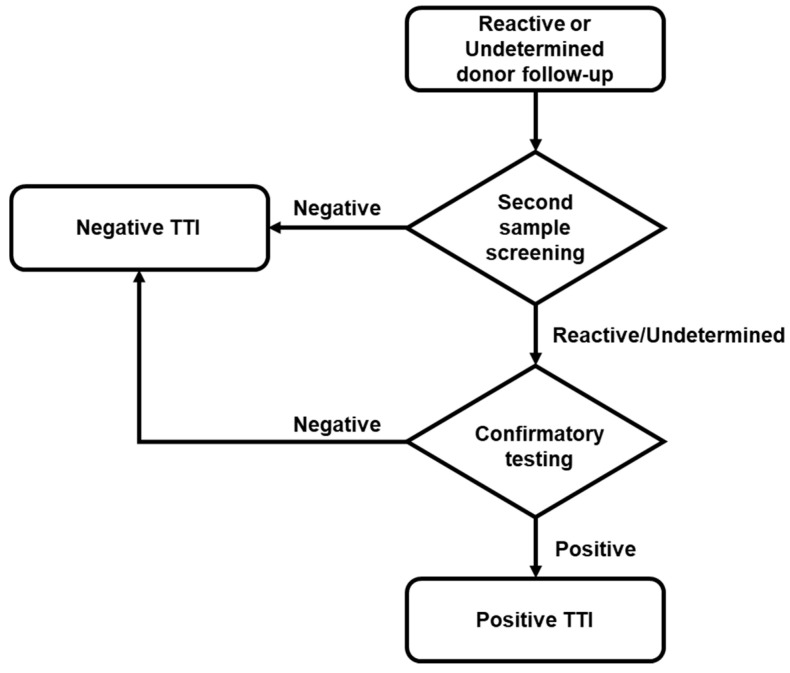
Flow chart of donor follow-up. According to regulations, localized reactive or indeterminate donors located underwent a second screening. Only when the screening of the second sample was reactive/indeterminate was the confirmatory test performed.

**Table 1 pathogens-10-00878-t001:** Demographic data of blood donors.

		Number	(%)
**Gender**	Female	26,399	32.8
	Male	53,992	67.2
	Total	80,391	100
**Age groups (years)**	18–30	33,938	43.4
	31–40	23,278	29.7
	41–50	15,006	19.2
	51–65	6054	7.7
	Total	78,276	100
**Location**	Metropolitan area	60,961	77.8
	Adjacent regions	17,356	22.2
	Total	78,317	100
**Blood group typing (ABO)**	O	48,175	59.9
	A	22,933	28.5
	A1	45	0.056
	A1B	8	0.010
	A2	23	0.029
	A2B	2	0.002
	AB	1599	2.0
	B	7606	9.5
	Total	80,391	100
**Rh (D antigen)**	Negative (−)	4624	5.8
	Positive (+)	75,767	94.2
	Total	80,391	100

**Table 2 pathogens-10-00878-t002:** Overview of HIV, HCV, and HBV seroreactive blood donors.

TTIs	Seroreactive	Percentage (%)	Rate	CI (95%)	Total
HIV	152	0.1891	18.91	15.90–21.91	80,391
HCV	385	0.4789	47.89	43.12–52.66	80,391
HBV	181	0.2251	22.51	19.24–2579	80,391

Seroprevalence is expressed as a percentage and as a rate, in cases per 10,000. TTIs, transfusion-transmissible infections; HIV, human immunodeficiency virus; HCV, hepatitis C virus; HBV, hepatitis B virus; CI, confidence interval (95%).

**Table 3 pathogens-10-00878-t003:** HIV, HCV, and HBV seroreactive blood donors classified by gender and age range.

		HIV			HCV			HBV			Total
		Seroreactive	Rate	CI (95%)	Seroreactive	Rate	CI (95%)	Seroreactive	Rate	CI (95%)	
**Gender**	Female	44	5.47	3.86–7.09	145	18.04	15.10–20.97	84	10.45	8.22–12.68	80,391
	Male	108	13.43	10.90–15.97	240	29.85	26.08–33.63	97	12.07	9.67–14.47	
**Age range**	18–30	69	8.81	6.74–10.89	124	15.84	13.06–18.63	78	9.96	7.75–12.18	78,276
	31–40	52	6.64	4.84–8.45	109	13.93	11.31–16.54	48	6.13	4.40–7.87	
	41–50	26	3.32	2.05–4.60	87	11.11	8.78–13.45	35	4.47	2.99–5.95	
	51–65	4	0.51	0.01–1.01	51	6.52	4.73–8.30	14	1.79	0.85–2.73	

Rates are expressed in cases per 10,000 and age range in years. HIV, human immunodeficiency virus; HCV, hepatitis C virus; HBV, hepatitis B virus; CI, confidence interval (95%).

**Table 4 pathogens-10-00878-t004:** Overview of seroprevalences based on NAT-reactive donors classified by seroreactive and seronegative (window periods) results.

TTIs	Seroreactive + NAT-Reactive	Percentage (%)	Rate	CI (95%)	Seronegative + NAT-Reactive	Rate	CI (95%)	Total (Positive-Confirmed Seroreactive)	Percentage (%)	Rate	CI (95%)
HIV	32	0.0398	3.98	2.60–5.36	1	0.12	0.00–0.37	33	0.041	4.10	2.70–5.51
HCV	45	0.056	5.6	3.96–7.23	4	0.5	0.01–0.99	49	0.061	6.10	4.39–7.80
HBV	20	0.0249	2.49	1.40–3.58	2	0.25	0.10–0.59	22	0.027	2.74	1.59–3.88

Rates are expressed in cases per 10,000. TTIs, transfusion-transmissible infections; HIV, human immunodeficiency virus; HCV, hepatitis C virus; HBV, hepatitis B virus; CI, confidence interval (95%).

## Data Availability

The data presented in this study are available on request from the corresponding author. The data are not publicly available due to privacy, and confidentiality, because they are sensitive information of the donors attended by the Central Blood Bank, in accordance with the applicable national and institutional regulations.

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
