# Peer review of "Retrospective Study of the Seroprevalence of HIV, HCV, and HBV in Blood Donors at a Blood Bank of Western Mexico"

_pathogens, 2021, doi:10.3390/pathogens10070878_

Round 1
Reviewer 1 Report
The submitted manuscript provides an overview of a detailed study on the blood donors of Western Mexico by screening their blood samples for serological markers and NAT. The study shows that HIV, HCV , and HBV seroprevalence is low in Western Mexico, and also reported differences in seroprevalence within the population based on gender and age. Overall, the manuscript emphasizes on the importance of methods to ensure safe blood transfusions, especially in underdeveloped and developing countries.
I have a few minor revisions:
- Please change the word "follow-up" to "followed-up" in the abstract.
- The acronym is PRT and not PTR in the abstract.
- In the introduction where the term NAT is introduced, could you please include a sentence on what NAT is and emphasize on why it is important to look at, in addition to serological markers, especially for HIV, HCV, and HBV?
- Line 179: Please change the letter W in the word western to uppercase
- Line 240: Correct the acronym PTR to PRT for the remainder of the manuscript. Also, please mention a sentence on what PRT is and how it is done.
- Your study main focuses on donors in metropolitan population. Can you speculate on how the results may differ if a broader population from various culturally, economically diverse areas are considered within Mexico? This may be a good addition to line 257 in the discussion section.
Author Response
Reviewer 1
minor revisions:
- Please change the word "follow-up" to "followed-up" in the abstract.
ANSWER: DONE
- The acronym is PRT and not PTR in the abstract.
ANSWER: DONE
- In the introduction where the term NAT is introduced, could you please include a sentence on what NAT is and emphasize on why it is important to look at, in addition to serological markers, especially for HIV, HCV, and HBV?
ANSWER: In agreement with the reviewer request, a couple of sentences has been included as follows: “The development, in the early 1990s, of molecular biology technologies such as polymerase chain reaction (PCR) and transcription-mediated amplification (TMA), allowed the implementation of Nucleic Acid Testing (NAT) for blood screening, in order to detect genetic material of HIV, HBV, and HCV, which has a paramount relevance for the reduction of the residual risk of window periods. Although only serological screening is mandatory in Mexico…” in addition: “NAT performance by using Procleix Ultrio Assay (GRIFOLS), as described in materials and methods.”.
- Line 179: Please change the letter W in the word western to uppercase
ANSWER: DONE
- Line 240: Correct the acronym PTR to PRT for the remainder of the manuscript. Also, please mention a sentence on what PRT is and how it is done.
ANSWER: In agreement with the reviewer suggestion, we add a sentence explaining what is PRT and very briefly in what it consists, as follows: “PRTs consist of technologies that must be implemented in blood banks, such as the use of U.V. or some chemicals, with the purpose of inactivate a broad variety of pathogens, such as: virus, bacteria and parasites, in blood components before transfusion”. In addition, because Central Blood Bank here studied did not use PRT, we insist to sensitize the institutions to investing in the use of PRT to decrease the risk of TTIs and increase the blood safety. To clarify this, we include a sentence as follows: “Currently, Central Blood Bank did not use PRT. In fact, in Mexico, only a few blood banks use PRTs, so it is still necessary to sensitize institutions about the advantages of investing in the use of PRTs with respect to the reduction of TTI infections and to also reduce the cost of patients care”.
- Your study main focuses on donors in metropolitan population. Can you speculate on how the results may differ if a broader population from various culturally, economically diverse areas are considered within Mexico? This may be a good addition to line 257 in the discussion section.
ANSWER: We thank the reviewer to point this aspect, in this regard, we included a sentence and modified the paragraph as follows: “One of the limitations of this study was that the population studied belongs to western Mexico, mainly the state of Jalisco, and there is not enough information about the rest of the country, and not all of it is current. For example, according to the few existing studies, HCV seroprevalence is higher in the states of Veracruz (1.1%, 2009) (Gulf of Mexico) [23] and Guanajuato (0.87%, 2020) (north-central) [24], compared with the seroprevalence reported in our study (0.48%, Table 2) and that of Yucatan (0.44%, 2006) (southeast) [25]. Therefore, a more robust and comprehensive study with population from different localities of the country is needed to have an overall picture of the seroprevalences of these TTIs in Mexico.”

Reviewer 2 Report
The article "Retrospective study of the seroprevalence of HIV, HCV, and HBV in blood donors at a Blood Bank of Western Mexico" by Garcia et al., describes the advantage of including NAT assay in addition to serological test from the blood donation to minimize TTI. The authors gave ample introduction about the existing problem and requirements of safety procedures to be followed. The article lack data from other demographical regions or different parts within Mexico as admitted by the authors.
- Though there is no association between the high number of reactive cases and gender, did the author observe any association of occupation/lifestyle? Studies have shown (https://www.jidc.org/index.php/journal/article/view/25771465), HCV risk was high among fishermen who made the donation.
- Table- headers to be corrected. abbreviations used has typo errors
- adding scientific/relevant info would provide good information to the readers. eg - instead of just mentioning serological test for HCV, samples were screened for hepatitis B surface antigen (HBsAg), antibodies to HIV 1 and 2.
-
All blood components from reactive or undetermined donors were immediately discarded.- please mention what guidelines were followed to safely discard these types of samples
Author Response
Reviewer 2
- Though there is no association between the high number of reactive cases and gender, did the author observe any association of occupation/lifestyle? Studies have shown (https://www.jidc.org/index.php/journal/article/view/25771465), HCV risk was high among fishermen who made the donation.
ANSWER: We thank the reviewer for the opportunity to emphasize this aspect. In agreement with the reviewer suggestion, we added, as supplementary material, a table with the Educational level and Occupation, with the available data from reactive donors. However, the educational level and occupation records of all donors were not available, so the association analysis was not carried out. In addition, a sentence has been included as follows: “When seroprevalences were classified by educational level, we observed that junior high school, high school and people with school degree, were the groups with higher seroprevalence of the three TTIs. Additionally, “Employee” was the occupational group with the highest seroprevalence in the three TTIs (Supplementary Table 1).”
- Table-headers to be corrected. abbreviations used has typo errors
ANSWER: DONE
- Adding scientific/relevant info would provide good information to the readers. eg - instead of just mentioning serological test for HCV, samples were screened for hepatitis B surface antigen (HBsAg), antibodies to HIV 1 and 2.
ANSWER: We thank the reviewer for point this relevant aspect, accordingly in the Materials and Methods, 2.3. Serological and NAT screening for HIV, HCV, and HBV, the paragraph was modified as follows: “Donor samples were obtained at the time of donation and then screened for serological HIV p24 antigen and anti-HIV-1 (Group M and Group O) and -HIV-2 antibodies; Hepatitis B surface antigen (HBsAg); and anti-HCV antibodies, according to national and institutional regulations (NOM-253-SSA1-2012) [20], and for NAT amplification at the same time”.
- All blood components from reactive or undetermined donors were immediately discarded.- please mention what guidelines were followed to safely discard these types of samples.
ANSWER: In the lines 335 and 336, the sentence has been modified as follows: “All blood components from reactive or undetermined donors were immediately discarded, according to national and institutional regulations (NOM-253-SSA1-2012)”
